# Stereopsis and Response Times between Collegiate Table Tennis Athletes and Non-Athletes

**DOI:** 10.3390/ijerph18126287

**Published:** 2021-06-10

**Authors:** Jiahn-Shing Lee, Shih-Tsung Chang, Li-Chuan Shieh, Ai-Yin Lim, Wei-Sheng Peng, Wei-Min Chen, Yen-Hsiu Liu, Lai-Chu See

**Affiliations:** 1Department of Ophthalmology, Chang Gung Memorial Hospital at Linkou, Guishan Dist., Taoyuan City 33305, Taiwan; leejsh@cgmh.org.tw; 2College of Medicine, Chang Gung University, Guishan Dist., Taoyuan City 33302, Taiwan; 3Office of Physical Education, Chung Yuan Christian University, Zhongli Dist., Taoyuan City 320314, Taiwan; cst87009@cycu.edu.tw; 4Graduate Institute of Sports Training, University of Taipei, Silin District, Taipei City 11153, Taiwan; shieh@utaipei.edu.tw; 5Department of Public Health, College of Medicine, Chang Gung University, Guishan Dist., Taoyuan City 33302, Taiwan; aiyinlim1@gmail.com (A.-Y.L.); weisheng@cgmh.org.tw (W.-S.P.); weiming@mail.cgu.edu.tw (W.-M.C.); 6Department of Physical Education, Chang Gung University, Guishan Dist., Taoyuan City 33302, Taiwan; 7Biostatistics Core Laboratory, Molecular Medicine Research Center, Chang Gung University, Guishan Dist., Taoyuan City 33302, Taiwan; 8Division of Rheumatology, Allergy and Immunology, Chang Gung Memorial Hospital, at Linkou, Guishan Dist., Taoyuan City 33305, Taiwan

**Keywords:** table tennis, athletes, stereopsis, eye–hand response time

## Abstract

Table tennis athletes and non-athletes potentially differ in stereopsis and eye–hand response times (RT), but whether stereopsis correlates with response time has scarcely been discussed. We aimed to compare stereopsis and RT between collegiate table tennis athletes and non-athletes and to examine the correlation between stereopsis and RT. From December 2016 to October 2019, table tennis athletes (*n* = 80) and non-athletes (*n* = 56) were recruited. Stereopsis was measured in four modes (A25, A50, R25, R50: approaching and receding directions at 25 mm/s or 50 mm/s). RT was measured with simple and choice tasks at zero, shoulder, and random distance. For stereopsis, the judged deviations were smaller during the approaching phases. Men had smaller A25 than women (*p* =0.012), whereas table tennis athletes showed smaller R25 and A50 than non-athletes (*p* = 0.03, 0.01, respectively). RT increased from simple to choice conditions and from zero to random, followed by shoulder distance. Men were significantly faster than women in choice tasks (*p* < 0.01). Table tennis athletes performed better in RT than the non-athletes (*p* < 0.05). No correlation was observed between stereopsis and RT (*r* = −0.01 to 0.12). In conclusion, athletes from table tennis sports showed better stereopsis and RT than non-athletes. Men had better stereopsis and RT than women. There was no correlation between stereopsis and RT.

## 1. Introduction

The number of studies on the role of vision science in sports has grown rapidly in the last decade, ever since academic research on sports vision blossomed in the 1980s [1]. The growing collaboration between athletic trainers, optometrists, and ophthalmologists in offering evidence-based sports vision services to athletes reflects the application of research findings to clinical practice [2]. The clinical application of sports vision is well established in the United States, but it is a relatively new area of optometric specialization in Asia and is gaining popularity with eye care practitioners and researchers [3,4].

A combination of visual skills is essential to guarantee satisfactory sports performance. The basic elements of sports vision include visual, perceptual, cognitive, and oculomotor abilities [4,5]. Sports vision encompasses comprehensive visual assessment, sports-specific enhancement of visual performance, and ocular protection for athletes, aiming to improve athletes’ performance [6,7]. Studies have demonstrated that athletes have better visual abilities than non-athletes [8] in terms of larger fields of peripheral vision [9,10,11], better dynamic visual acuity [12,13], and more accurate depth perception [14].

Visual reaction time significantly affects athletes’ perceptive skills. Response time, which refers to the latency until a person moves in response to a stimulus, is a critical metric in most sports. Response time requires intact sensory skills, decision processing, and motor performance [15]. A ‘simple’ situation involves only one type of stimulus, and a ‘choice’ situation challenges an individual with more than one type of stimuli. Choice response time involves information processing, including the four identified stages of stimulus coding, stimulus–stimulus translation, stimulus–response translation, and response selection [16]. Our previous research compared the visual response time between karate athletes and non-athletes, revealing a similar finding where athletes demonstrated superior simple and choice response time [17]. Another study of ours showed that six weeks of visuomotor training improved visual response time in collegiate karate athletes [18].

Stereopsis (otherwise known as binocular depth perception) greatly affects sports performance. Stereopsis is traditionally considered to be the threshold measure of how well an individual can interpret binocular disparity as perceived depth by determining the spatial correlation of points projected onto the retina [19]. As a physical object moves toward or away from an individual, monocular and binocular cues are needed to determine depth change. Any point that falls in front of fixation provides crossed disparity; that is, these points are projected on the temporal retina of both eyes. This is a binocular disparity, as the corresponding point to the temporal retina of one eye is the nasal point of the other eye. This advanced visual function based on binocularity disparity is especially important in skill-specific sports [20,21].

Table tennis is one of the fastest ball games because of the high-speed ball traveling in the short distance between two players [22]. The player requires fast RT and good stereopsis to process the visual information about the approaching ball. Previous studies have shown that young, professional table tennis players show better stereopsis than age-matched students [23]. Visual RT is a critical factor to determine the performance level of table tennis athletes [24,25].

Current practice in racket sports emphasizes visual training in athletes’ routine workout. While response time and other visual abilities have been widely studied, stereopsis in sportsmen has not yet been investigated. Even though stereopsis has been commonly tested in ophthalmology and optometry practice, we found little evidence of it relating to the sporting context. 

In this study, we first compared stereopsis and eye–hand response time between collegiate table tennis athletes and non-athletes. Next, we explored the relationship between stereopsis and eye–hand response time among college students (collegiate table tennis athletes and non-athletes). 

## 2. Materials and Methods

### 2.1. Study Subjects

From December 2016 to October 2019, we recruited table tennis athletes (*n* = 80, male: 54.41%, mean age: 22.01 ± 2.72 years, BMI: 21.78 ± 3.46 kg/m^2^) from a sports university (University of Taipei) and non-athletes (*n* = 56, male: 44.64%, mean age: 22.73 ± 3.49 years, BMI: 22.07 ± 4.27 kg/m^2^) from a non-sports university (Chang Gung University). The inclusion criteria were university students with no significant medical conditions and no injury or surgery in the past three months. The exclusion criteria were self-reported color blindness, strabismus, known irreversible eye diseases, and impaired depth perception. We obtained approval by the Institutional Review Board of the Chang Gung Medical Foundation (201600729A3, 201601220B0, 201701432B0). All subjects signed an informed consent form after being informed about the nature and possible consequences of the study. Parental consent was obtained if the subjects were below 20 years old.

Because it is well known that men have a faster response time than women do [26,27,28], we stratified demographic characteristics by sex. Among male subjects, age, BMI, right-handedness, game hours, and PC hours were similar between the two study groups. Among female subjects, there was no significant difference in BMI, right-handedness, and game hours between the two study groups. However, older age (*p* = 0.002) and spending more PC hours (*p* = 0.005) was observed more in the non-athletes than the table tennis athletes (Table 1). 

### 2.2. Measurements

All eligible participants received basic visual tests, including measurement of visual acuity (VA) and refractive status, to ensure that their bare or corrected vision was normal during the stereopsis and RT tests. 

#### 2.2.1. Stereopsis

Stereopsis was measured using an electric DP tester (Takei Kiki Kogyo Co. Ltd., Niigata, Japan) in a room with illuminance between 520 and 710 Lux (a comfortable lighting range, not too dark and not too bright). We have previously used this DP tester to measure stereopsis [29]. The DP tester is a modified Howard–Dolman apparatus, further enhanced by enclosing the rods in housing with two 15 W fluorescent lamps and providing a uniform and diffused lighting system. There are three identical rods inside the tester. The rods on either side are immovable, while the middle rod can be moved to be in front of (approaching the participant) or behind (receding from the participant) the other two static rods. This motorized apparatus allows the examiner to move the rod at two speeds: 25 mm/s and 50 mm/s. 

Subjects were seated 2.5 m away from the apparatus, facing the open window (18 cm × 9 cm) of the apparatus, and tested with their best-corrected vision. The examiner controlled the direction and movement speed of the movable rod by two buttons at the side of the box. Subjects were instructed to fixate on the middle of the 2 static rods and not watch the movable rod. Subjects were asked to press a wired button connected to the apparatus to stop the moving rod when they judged that all three rods were in alignment. 

The amount of absolute deviation in cm was determined based on the alignment of the middle rod and static rods, whereby the smaller the judged deviation (cm), the better the stereopsis. Subjects practiced the 4 modes of stereopsis, i.e., approaching at 25 mm/s (A25), receding at 25 mm/s (R25), approaching at 50 mm/s (A50), and receding at 50 mm/s (R50), 3 times. Subjects took the test of stereopsis 3 times for each measurement. The largest value for each subject was eliminated, and the average of the remaining 2 values was used for analysis. 

#### 2.2.2. Eye–Hand Response Time

Eye–hand response time was measured using the FitLight Trainer system manufactured by FitLight Sports Corp., Canada, a wireless light system composed of eight RGB LED-powered lights and controlled by a tablet. The lights were used as targets for the subject to deactivate per the examination routine. The measuring protocol on eye–hand response time has been used in our previous studies [17]. 

Subjects were examined with their best-corrected vision. They stood facing the wall, an arm’s length away, and were asked to raise the tested arm to chest level in front of their chest with their palm gently touching the wall (beginning position). Each subject was instructed to deactivate lights as quickly as possible by placing the tested hand in close proximity (within 10 cm) to a flashing light. After deactivating the light, they had to return the tested arm to the beginning position. The measurement was configured by the tablet controller, and 30 s of rest was provided between each measurement. 

Two simple tasks and three choice tasks of eye–hand response time (ms) were measured. Simple tasks were programmed with a green light (Go task), while choice tasks comprised green and red lights (Go/No-Go task). Lights mounted on the wall were positioned at three different distances from the subject’s hand: zero, shoulder, and random distances. For zero distance, one light disc was mounted on the wall, and the subject’s tested hand was positioned right next to the disc. For shoulder distance, the subject’s tested hand was positioned a shoulder’s width away from the light disc. For random distance, eight light discs were mounted on the wall, and the subject’s tested hand was positioned at the center of the discs. Five bouts were administered for simple response time measured at zero distance (SRT_zero) and shoulder distance (SRT_shoulder). A total of 10 bouts were administered for choice response time measured at zero distance (CRT_zero) and shoulder distance (CRT_shoulder), while 30 bouts were administered for choice response time measured at a random distance (CRT_random) in consideration of the higher variation of response time in more difficult conditions.

As the mean is easily distorted by outliers due to inattention or external disturbances, we used a trimmed mean by eliminating the highest and lowest values of each measurement for every participant.

### 2.3. Statistical Analysis

SAS 9.4 software was used to analyze the data. All continuous variables have been summarized in mean and standard deviation in the text and in the tables. Independent *t*-tests or Chi-square tests were conducted to compare demographic variables between table tennis athletes and non-athletes where appropriate. Two-way analysis of variance (ANOVA), one form of the generalized linear model (GLM) [30], was performed to compare stereopsis or response time between sexes (male vs. female) and sports groups (table tennis athletes vs. non-athletes). Residual of diagnostic plots were created to check the assumption of GLM, including linearity, normality, and homoscedasticity [31]. No particular violation of the assumptions was observed in our data. The procedure of two-way ANOVA was as follows: (1) Interaction between sex and sports group was examined; (2) if the interaction was significant, the main effect of sex and the main effect of sports group were considered meaningless because the difference in stereopsis and response time (dependent variable) between sex depended on sports groups, or vice versa; (3) if the interaction was insignificant, the main effect of sex and sports group was meaningful. For instance, when the main effect of sex was significant, we could simply say that the dependent variable differed between sexes without worrying about the sports group to which they belonged [32]. When significance was reached, we used Tukey’s multiple comparisons to locate which group is different from which group. In Tukey’s multiple comparisons, the mean square error (MSE) was from the two-way ANOVA with the main effects of sex, sports group, and interaction between sex and sports group [33]. Because *p* values are highly affected by the sample size [34], we also reported the effect size. For two-way ANOVA, semipartial eta^2^ (η^2^) (a proportion of total variation “partialed out” of other covariates in the dependent variable) is used. Cohen (1988) designated effect sizes as small (0.1), medium (0.25), and large (≥0.4) [35]. The Pearson correlation coefficient was used to indicate the linear relationship between stereopsis and response time. Because the multiple measures in this study may suffer from the high type I error, we used a simple sequential Bonferroni-type procedure to deal with the multiple measures issue. The sequential Bonferroni-type procedure has been proved to control the false discovery rate (= type I error) in independent test statistics and to contribute to power gain in multiple measures. The simple sequential Bonferroni-type procedure includes (1) ranking the *p* values of multiple tests from smallest to largest as *p*_(i)_ and (2) determining whether *p*_(i)_ reaches significance if *p*_(i)_ ≤ i/k × Q, where k is the total number of multiple tests and Q is the overall significant level (=0.05) [36]. The significance level in this study was set at 0.05. 

## 3. Results

For stereopsis, the judged deviations were smaller during the approaching phases. Interaction between sex and sports group was insignificant for R25 (η^2^ = 0.0011, *p* = 0.69), A25 (η^2^ = 0.0005, *p* = 0.79), R50 (η^2^ = 0.0031, *p* = 0.52), and A50 (η^2^ = 0.0004, *p* = 0.82). The significant main effect of sex was only seen for A25 (male: 1.02 ± 0.81 cm vs. female: 1.56 ± 1.49 cm, 95% CI of the difference = 0.15 to 0.95, η^2^ = 0.0447, *p* = 0.012) (Figure 1). The significant main effect of sports group was seen for R25 (table tennis athletes: 1.09 ± 0.92 cm vs. non-athletes: 1.65 ± 1.8 cm, 95% CI of the difference = 0.11 to 1.01, η^2^ = 0.0394, *p* = 0.03) and A50 (table tennis athletes: 0.89 ± 0.84 cm vs. non-athletes: 1.41 ± 1.3 cm, 95% CI of the difference = 0.16 to 0.89, η^2^ = 0.0483, *p* = 0.01) (Figure 2).

For eye–hand response time, simple and zero distance tasks required shorter amounts of time than choice and shoulder/random distance tasks. On average, the participants took 366.55 ± 68.06 ms for SRT_zero, 381.82 ± 60.37 ms for SRT_shoulder, 422.07 ± 67.89 ms for CRT_zero, 488.23 ± 83.56 ms for CRT_shoulder, and 484.28 ± 79.11 ms for CRT_random. There was no significant interaction between sex and sports group (η^2^= 0.0013, *p* = 0.66 for SRT_zero; η^2^ = 0.0008, *p* = 0.74 for SRT_shoulder; η^2^= 0.0008, *p* = 0.74 for CRT_zero; η^2^ = 0.0064, *p* = 0.39 for CRT_shoulder, and η^2^ = 0.0002, *p* = 0.88 for CRT_random). A significant main effect of sex was seen on most RT, except for SRT_shoulder. Men were significantly faster than women in SRT_zero (355.3 ± 70.86 ms vs. 380 ± 62.48 ms, 95% CI of the difference = 1.84 to 47.59 ms, η^2^ = 0.0169, *p* = 0.03), CRT_zero (408.3 ± 64.4 ms vs. 438.5 ± 68.78 ms, 95% CI of the difference = 7.63 to 52.88 ms, η^2^ = 0.0432, *p* = 0.01), CRT_shoulder (473.4 ± 77.3 ms vs. 511.1 ± 88.49 ms, 95% CI of the difference = 6.32 to 69.09 ms, η^2^ = 0.0297, *p* = 0.02), CRT_random (471.4 ± 78.71 ms vs. 504.1 ± 76.41 ms, 95% CI of the difference = 2.88 to 62.55 ms, η^2^ = 0.0286, *p* = 0.03) (Figure 3). The significant main effect of sports group was seen in simple tasks and CRT_random. The table tennis group was significantly faster than the non-athlete group in SRT_zero (350.7 ± 58.72 ms vs. 389.2 ± 74.33 ms, 95% CI of the difference = 15.96 to 61.15 ms, η^2^ = 0.0636, *p* = 0.001), SRT_shoulder (370.4 ± 54.92 ms vs. 395.7 ± 65.05 ms, 95% CI of the difference = 4.79 to 45.65 ms, η^2^ = 0.0335, *p* = 0.02), and CRT_random (466.6 ± 65.1 ms vs. 520.1 ± 92.86 ms, 95% CI of the difference = 23.44 to 83.41 ms, η^2^ = 0.0898, *p* = 0.003) (Figure 4).

The correlation coefficients between stereopsis and eye–hand response time ranged from −0.07 to 0.18. According to the sequential Bonferroni-type procedure, none of these correlation coefficients reached statistical significance (Table 2).

## 4. Discussion

Our findings summarize the mean of stereopsis and eye–hand response time between young men and women, and between table tennis athletes and non-athletes. For stereopsis, the judged deviations were smaller during the approaching phases. Men had better A25 than women, whereas table tennis athletes showed better stereopsis of R25 and A50 than non-athletes. However, the effect size of sex on A25 and the effect size of sports group on R25 and A50 are very small (0.0477, 0.0394, 0.0483, respectively). For eye–hand response time, our results showed that response time increased from simple to choice conditions and from zero to random, followed by shoulder distance. Men reacted significantly faster than women in simple_zero and choice tasks. Additionally, table tennis athletes performed better in the response time task than the non-athletes. Again, the effect size of sex and sports group on response time is very small (0.0169–0.0898). No correlation between stereopsis and response time was observed. 

For stereopsis, we saw that the judged deviations were smaller during the approaching phases. The better performance of stereopsis for the approaching mode than receding mode can be explained by Perrone’s work [37]. Perrone proposed that there are two separate systems for processing motion toward the eye and motion away from the eye, because backward locomotion is usually a non-threatening action to humans and our visual system is thus less developed to deal with receding motion. 

In this study, we saw better stereopsis of A25 in men than in women. Whether this is related to the shorter reaction times of men in the sample or to a sex difference in stereoacuity deserves further study. To the best of our knowledge, only the study by Zaroff et al. compared stereopsis between men and women among a wide age range of subjects (from 15 to 79 years old) [38], and they revealed no statistically significant sex differences in optimal stereoacuity or in age-related influences on stereoacuity. It is noteworthy that Zaroff used random dot stereograms, not a real depth perception, for measurements of stereopsis. Women have smaller interocular distances (IODs), and thus, a given distance in cm causes a smaller angular disparity. Studies measuring stereoacuity in terms of angular disparity have typically found no sex difference. If stereoacuity is indeed equal in terms of angular disparity, women would be expected to score slightly worse in terms of real depth in cm. However, the literature in this area has not revealed the expected correlations between IOD and stereopsis. 

In this study, table tennis athletes showed better stereopsis of A50 and R25 than non-athletes. The sports involvement of table tennis athletes, particularly in reacting and discriminating the quick approaching ball, potentially explains the superiority (compared to the non-athletes) of stereopsis at A50 and R25 in our study. In table tennis, projectile flight speed is as high as 170 km/h, which is much faster than in our test. However, we could not conclude that there was a translation of on-field racket stroke in a game to the response in stopping the rod while tested on the Howard–Dolman test. There are not many studies which evaluate athletes’ stereoscopic function using the Howard–Dolman apparatus [39]. Motz et al. (2017) examined the effects of daily exercises in improving stereopsis, peripheral vision, and visual reaction time, in which a Howard–Dolman apparatus was used to measure the stereopsis of 45 children aged 9–14 participating in recreational sports teams. Their results showed the distance measured by the Howard–Dolman apparatus improved after 4 weeks of training. However, details of the result, such as the definition of improvement, were not mentioned. 

Our result showed that RT increased from simple to choice conditions and from zero to random, followed by shoulder distance. Our finding is in line with Hick’s law [40], which reports that a person requires more time to make a decision as a result of the possible choices they face. Different distances from the initial point to the ending point during RT measurement resulted in different duration. Zero distance in our study was presumed by placing a palm right next to the stimulus. Therefore, only minimum motor action is required for this condition, while shoulder distance required more motor action from bigger muscle groups. Among the three distances tested for choice RT, random distance appeared to be shorter than shoulder distance as the distance between closer and farther lights was averaged out.

Our study also showed that men had a significantly faster RT than women. This result is in line with a couple of studies which investigated factors associated with RT among young adults [26,27,41]. A significant sex difference in response time is well known. Adam et al. [26] reported that men have a faster response time than women because they use a specific information processing strategy that differs from the strategy used by women.

Our results showed table tennis athletes perform better in the response time task than non-athletes, which is in accordance with our previous study on karate athletes [17]. Laby et al. (2018) showed a significant relationship between the eye–hand visual–motor reaction time and batting performance among 450 professional baseball players [42]. The better response time among athletes than non-athletes can be explained by the fact that skillful athletes generally possess better skill, accuracy, and spatial–temporal constraints in the realm of visual information acquisition and processing. 

In this study, there was no significant correlation between stereopsis and response time in college students and table tennis athletes. Blake et al. reported that binocular response time was consistently faster than monocular response time [43]. Since our response time was already measured under binocular vision, the correlation between stereopsis and response time under binocular vision is minimal. We expected RT to be significantly associated with stereopsis, but our finding could not support this hypothesis. Different neural pathways of stereopsis and RT could partly explain this insignificant relationship. Our RT tasks measured both sensory and motor reactions. Stereopsis is a central processing task primarily on the visual pathway [44], whereas RT involves sensory input, information processing, and motor output [15]. Sensory and motor processing are attributed mainly to the somatic nervous system. A comparison of stereopsis and RT is less relevant than directly comparing the central processing component of the RT task with stereopsis. Evaluating brain activity while testing response time may be a solution. Recent advanced studies have used electroencephalographs to examine time delay in response, indicating brain activity [45,46].

Researchers have argued that typical static stereoacuity tests are insufficient to reveal the full picture of stereopsis in sports vision [47]. Although the Howard–Dolman apparatus tests dynamic stereopsis, the device tests along the midline and at a much slower speed compared to that normally used in table tennis. We believe that the Howard–Dolman apparatus is not an ideal test for athletes’ stereopsis. Paulus et al. suggested an extended stereopsis evaluation for athletes, which combined stationary and moving stimuli to cover static and dynamic stereopsis. Their study used four alternative forced choice tests displayed on a 3D TV. However, their measurement is not in line with ours, given the combination of stereopsis and RT into a 3D RT task. Their finding is congruent with ours, where athletes have a better choice RT but not stereopsis task than non-athletes [47]. Although static stereopsis is consistently shown in literature as superior in athletes [23,48], dynamic stereopsis in athletes remains inconclusive. A standard measurement of dynamic stereopsis for athletes is urgently needed to reach a consensus. 

There are limitations to this study that warrant attention. The first regards the subjects, and it is that athletes were recruited from a sports university and non-athletes from a non-sports university, resulting in potentially differing demographic characteristics that were not collected in this study. In fact, the non-athlete group participated in this study in 2016–2017, while the racquet sports group participated in 2018–2019; this was because three different grants were involved. We believe that this time difference in the two study groups should not affect the result. The second limitation regards stereopsis, and it is that the unit of stereopsis used in this study was cm, rather than arc seconds, because interpupillary distances were not measured in this study. In future studies, measuring interpupillary distances is highly encouraged. The third limitation is that we did not analyze the participants’ heterophorias and fusional vergence reserves (convergence and divergence) at far vision to evaluate their effect on fine stereopsis because it was beyond our study’s scope. The effects of horizontal heterophoria on Howard–Dolman stereopsis have been investigated before, and it was found that exophoria showed no systematic and statistically valid effect on stereopsis, but esophoria did show an effect [49]. The prevalence of heterophoria at far distance was estimated as 3.82% having exophoria, with only 0.33% having esophoria and 0% having vertical heterophoria in a Chinese population [50]. Owing to the very low prevalence of esophoria and vertical heterophoria, we believe that their effect on fine stereopsis in our results should not be significant. The fourth limitation regards the response time, and it is that the latency of the measurement tool (FitLight Trainer system) was unknown. The response time obtained in this study could not justify the effects of the movement strategy to deactivate the light. Finally, we recognized that the number of PC hours was smaller in the table tennis sports group than in the non-athlete group. However, we put game hours as a covariate in the analysis of covariance (ANCOVA) and saw similar results (data not shown).

## 5. Conclusions

Table tennis athletes showed better stereopsis and eye–hand response time than non-athletes. Men had better stereopsis and RT than women. There was an insignificant correlation between stereopsis and eye–hand response time.

## Figures and Tables

**Figure 1 ijerph-18-06287-f001:**
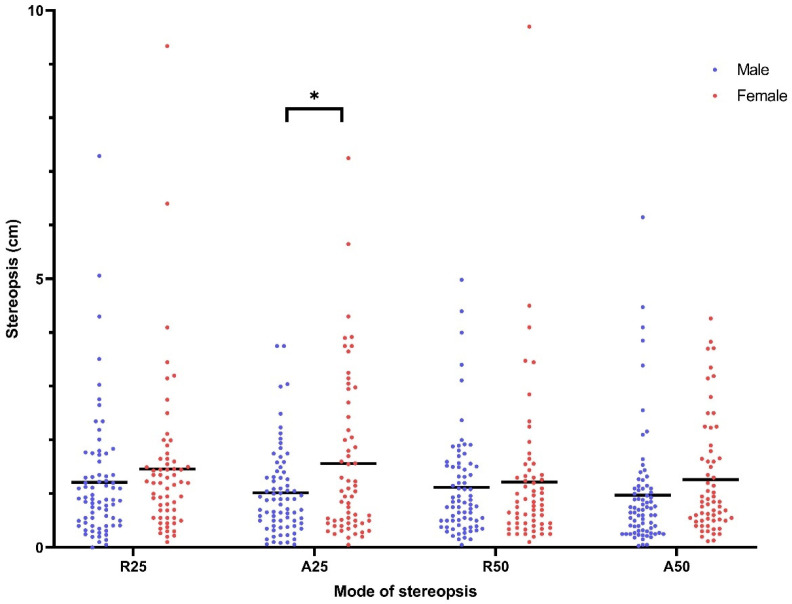
Stereopsis of male and female college students measured in four modes of stereopsis. (*: statistical significant between males and females).

**Figure 2 ijerph-18-06287-f002:**
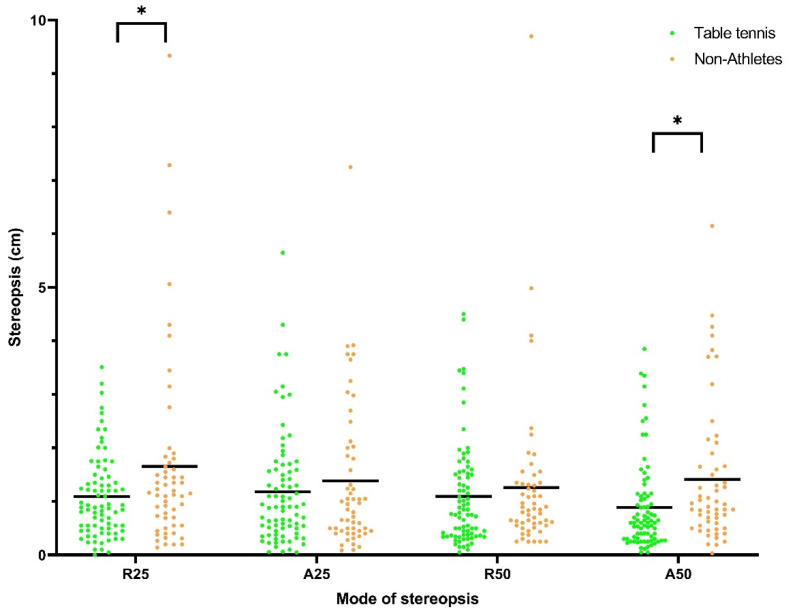
Stereopsis of table tennis athletes and non-athletes measured in four modes of stereopsis. (*: statistical significant between table tennis athletes and non-athletes).

**Figure 3 ijerph-18-06287-f003:**
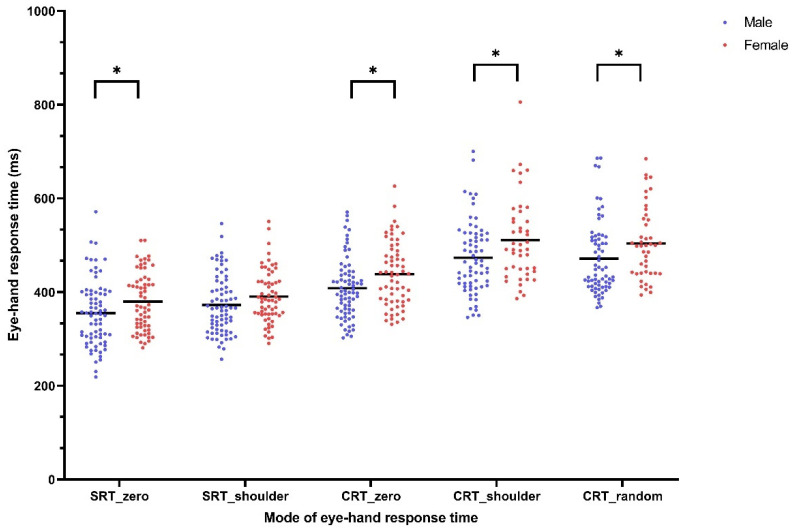
Eye–hand response time of male and female college students measured in five modes of eye–hand response time. (*: statistical significant between males and females).

**Figure 4 ijerph-18-06287-f004:**
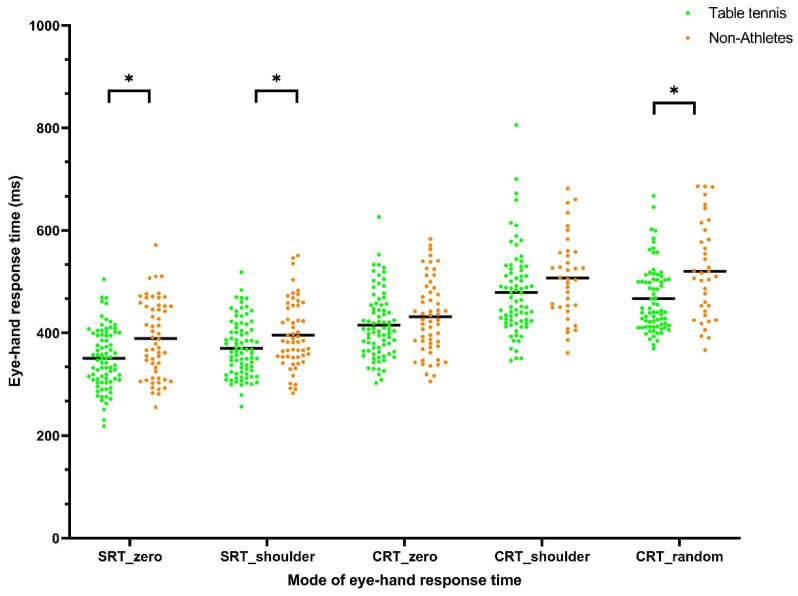
Eye–hand response time of table tennis athletes and non-athletes measured in five modes of eye–hand response time. (*: statistical significant between table tennis athletes and non-athletes).

**Table 1 ijerph-18-06287-t001:** Demographic characteristics between table tennis athletes and non-athletes by sex (*n* = 240).

Variables	Total (*n* = 136)	Table Tennis Athletes (*n* = 80)	Non-Athletes (*n* = 56)	*p*
Male (*n*)		49	25	
Age (mean ± SD, years)	22.07 ± 2.62	21.49 ± 2.1	22.32 ± 3.45	0.62 ^1^
BMI (mean ± SD, kg/m^2^)	21.96 ± 3.28	21.76 ± 2.73	22.34 ± 4.2	0.54 ^1^
Right-handedness	62 (83.78%)	41 (83.67%)	21 (84.0%)	0.76 ^2^
PC hours (mean ± SD)	20.8 ± 24.12	18.94 ± 23.36	24.38 ± 25.62	0.36 ^1^
Game hours (mean ± SD)	15.35 ± 24.36	13.13 ± 17.5	19.62 ± 33.94	0.38 ^1^
Female (*n*)		31	31	
Age (mean ± SD, years)	21.95 ± 2.86	20.83 ± 1.22	23.06 ± 3.54	0.002 ^1^
BMI (mean ± SD, kg/m^2^)	21.56 ± 3.69	21.26 ± 2.84	21.85 ± 4.39	0.53 ^1^
Right-handedness	56 (90.32%)	27 (87.1%)	29 (93.5%)	0.67 ^2^
PC hours (mean ± SD)	14.66 ± 17.2	8.38 ± 9.68	20.53 ± 20.51	0.005 ^1^
Game hours (mean ± SD)	8.01 ± 14.74	9.5 ± 16.84	6.56 ± 12.49	0.44 ^1^

Remarks: PC hours refers to hours spent using the computer in a week; game hours refers to hours spent playing computer games in a week. ^1^ Independent *t*-test; ^2^ Chi-square test.

**Table 2 ijerph-18-06287-t002:** Correlation (correlation coefficient, *r*) between stereopsis and eye–hand response time (*n* = 136).

	SRT_Zero	SRT_Shoulder	CRT_Zero	CRT_Shoulder	CRT_Random
R25	0.09	0.1	0.02	0.001	0.03
A25	0.02	−0.04	−0.03	−0.01	−0.07
R50	0.12	0.13	0.09	0.18	0.14
A50	0.08	0.02	0.07	0.09	0.05

Remarks: A25, A50, R25, R50: approaching and receding directions at 25 mm/s or 50 mm/s; SRT_zero, SRT_shoulder: simple response time at zero distance or at shoulder distance; CRT_zero, CRT_shoulder, CRT_random: choice response time at zero distance, at shoulder distance or at random distance; according to the sequential Bonferroni-type procedure, none of these correlation coefficients reached statistical significance because the smallest *p* was 0.005 (>1/20 × 0.05 = 0.0025) for the correlation coefficient between R50 and CRT_random (*r* = 0.18).

## Data Availability

The data presented in this study are available on request from the corresponding author.

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
