# Peer review of "Stereopsis and Response Times between Collegiate Table Tennis Athletes and Non-Athletes"

_ijerph, 2021, doi:10.3390/ijerph18126287_

Round 1

Reviewer 1 Report

As I indicated at the time, I believe that this type of research provides additional information in the area of sports vision, and I find your research very suggestive.  I thank the authors who have responded and improved in this new version of the manuscript each of the points that I had suggested, so I congratulate you on it. I only have two observations that would have to be considered for the final publication:

  • I see that the ambient illuminance range (line 116) is very wide, such as 520-710 lux. It would be plausible to clarify the reason for this, if this variability is due to the fact that the testing was carried out in different areas of the room, etc. I think that the really important thing is to know the illuminance in the area where the DT Tester stimuli are located, and I am not sure that the luminance range is that wide in that area.
  • I suggest that the P-value throughout the results section be specified in italics and lowercase, that is, like this: p

Author Response

As I indicated at the time, I believe that this type of research provides additional information in the area of sports vision, and I find your research very suggestive.  I thank the authors who have responded and improved in this new version of the manuscript each of the points that I had suggested, so I congratulate you on it. I only have two observations that would have to be considered for the final publication:

  • I see that the ambient illuminance range (line 116) is very wide, such as 520-710 lux. It would be plausible to clarify the reason for this, if this variability is due to the fact that the testing was carried out in different areas of the room, etc. I think that the really important thing is to know the illuminance in the area where the DT Tester stimuli are located, and I am not sure that the luminance range is that wide in that area.

Reply: The wide range of ambient illuminance (520-710 lux) in the test room is within the comfortable lighting for a room light (not too dark and not too bright). There are two 15 W fluorescent lamps inside the DT tester as a lighting source. I don’t think the illuminance in the test room will affect the measure of stereopsis. We revised the text as “Stereopsis was measured using an electric DP tester (Takei Kiki Kogyo Co. Ltd., Japan) in a room with illuminance between 520-710 Lux (a comfortable lighting range, not too dark and not too bright). We had used this DP tester to measure stereopsis previously[29]. The DP tester is a modified Howard-Dolman apparatus, further enhanced by enclosing the rods in housing with two 15 W fluorescent lamps and providing a uniform and diffused lighting system. “ (line 115-120, this re-submission)

  • I suggest that the P-value throughout the results section be specified in italics and lowercase, that is, like this: p

Reply: all the P-value throughout the results section is revised in italics and lowercase.

Reviewer 2 Report

The authors have tried to justify the suggestions made.
They have provided information on methodological decision-making.
In spite of having reduced the sample by eliminating several population groups, the study is now more adequate, since residual samples from other sports are not used.
The manuscript can be published

Author Response

The authors have tried to justify the suggestions made.
They have provided information on methodological decision-making.
In spite of having reduced the sample by eliminating several population groups, the study is now more adequate, since residual samples from other sports are not used.
The manuscript can be published.

Reply: Thank you for your support.